# Mental Health and Relational Needs of Cambodian Refugees after Four Decades of Resettlement in the United States: An Ethnographic Needs Assessment

**DOI:** 10.3390/bs14070535

**Published:** 2024-06-26

**Authors:** Chansophal Mak, Elizabeth Wieling

**Affiliations:** 1Department of Family Social Science, College of Education and Human Development, University of Minnesota, Minneapolis, MN 55455, USA; 2Department of Human Development and Family Science, College of Family and Consumer Sciences, University of Georgia, Athens, GA 30602, USA; ewieling@uga.edu

**Keywords:** mental health, family relationships, ethnographic needs assessment, Cambodian refugees, traumatic stress, intergenerational transmission of trauma, post-resettlement stress

## Abstract

The United States has a long history of welcoming refugees fleeing persecution, organized violence, and war. However, the welcome often does not come with adequate immigration infrastructure support necessary to rebuild life and promote family well-being. Approximately 157,000 Cambodians were accepted to resettle in the U.S. between 1975 and 1994 due to the countrywide genocide. Upon resettlement, Cambodians were placed in impoverished neighborhoods with little resources to heal and rebuild. The purpose of this study, grounded in a Human Ecological Model and guided by Critical Ethnography principles, was to conduct a formal needs assessment of Cambodian refugee families across the United States. Eighteen professionals were interviewed virtually in Khmer and/or English. The data were analyzed using the Developmental Research Sequence. The results emphasized a critical need to address mental health complications resulting from untreated mental health disorders such as posttraumatic stress, depression, anxiety-related disorders, and complicated grief, across generations. Severe disruptions in family relationships (i.e., parent–child and couple relationships) were also reported along with substance abuse in the absence of access to culturally responsive mental health treatments. Findings suggest the need for culturally tailored multilevel interventions to effectively address mental health and relational challenges of multigenerational Cambodian families.

## 1. Introduction

Complicated individual psychological trauma and its intergenerational transmission effects among war-torn populations is a global mental health problem [1,2]. Organized violence, social injustice, and natural disasters force people to flee from their birth countries in search of safety elsewhere [3]. It was reported that by May 2023, approximately 110 million people globally were forcibly displaced, including 41 percent of children among them [3]. 

Forcibly displaced populations face multiple adversities, such as financial hardship and limited access to social security and health care services [4]. Aspects of forced displacement usually involve exposure to traumatic events such as witnessing the death of family members, loss of home and belongings, family separation, harsh labor, and persecution [5]. Despite intergenerational resilience in the face of extreme adversity, experiencing these traumatic events severely affects refugees’ mental health and threatens their ability to connect to their family and community unless critical support systems are ensured [6]. 

The United States (U.S.) has a history of admitting Indochina war survivors from Southeast Asia in the 1970s [7], yet the U.S. does not have a good record of providing the necessary support to help immigrants who experienced horrific traumas to heal and rebuild at post-resettlement. Cambodian refugees arrived in the U.S. along with other Indochina war survivors after they experienced the Cambodian genocide, also known as the Khmer Rouge or the Pol Pot regime (1975–1979) [8,9]. The Khmer Rouge, sponsored by the communist Chinese government, ruled Cambodia using mass violence as a method to transform Cambodian society [9]. About two million citizens in Phnom Penh, the capital city of Cambodia, were forced to move to assigned rural areas [9]. Significantly, Cambodians experienced extreme disruption of family dynamics as family members were dispersed and placed in different work locations according to their age groups and genders [9]. Adults were forced to work excessively on the rice field, while children were placed in different work camps and brainwashed. By the end of the regime, about half of the Cambodian population perished due to political executions, exhaustion, famine, and illnesses [9,10].

As the Khmer Rouge regime ended in January 1979, the survivors migrated within the country in search of their separated family members [11]. Simultaneously, many Cambodians risked their lives crossing the border to refugee camps in Thailand and continued to live in harsh conditions (i.e., limited food supplies, lack of support systems, untreated psychological trauma, and physical ailments) for years before obtaining admission to the U.S. for resettlement [12,13].

Approximately 157,000 Cambodian refugees were resettled in the U.S. by the 1990s along with other Indochina war survivors [11]. Cambodian refugees continued to be exposed to traumatic stressors in addition to post-resettlement stressors as they had to rebuild their family from scratch along with limited start-up support for newly resettled refugees in the U.S. [14]. Additionally, they faced unemployment due to limited education and barriers in language and transportation, as many of them were preliterate peasants [15,16]. Even though some were able to obtain entry-level jobs, many lived on disability funding [11]. Most importantly, posttraumatic stress disorder (PTSD) and complicated grief due to severe trauma exposures and losses during the genocide remain under-documented and untreated [17,18]. Therefore, the ongoing consequences of poor mental health and disrupted family relationships persist within the Cambodian communities across the U.S. [19].

### Mental Health and Relational Consequences of Trauma Exposure among Cambodian Refugees

This section employs an ecological systems approach to discuss mental health and relational disruptions in family life related to war and resettlement at each system level.

In the individual subsystem, refugees escaping organized violence are prone to mental health complications such as PTSD, depression, adjustment disorders, and grief-related disorders [20,21]. Similarly, Cambodian refugees experienced life-threatening events daily during the genocide, in refugee camps, and post-resettlement in the U.S. [22,23]. A host of psychological comorbidities, physical illnesses, substance abuse, poverty, low levels of education and social capital has also been identified at the individual level.

In the family subsystem, family separation and death of family members during the war and migration severely disrupt the family process of refugee communities [24,25]. Similarly, the genocide left Cambodian families with tremendously painful memories and disrupted family structure and process due to the death of significant family members [9]. After resettlement, they were trapped in survival mode and had no time to process their emotional pain and grief. Additionally, they also experienced ambiguous losses within their families that frequently manifested in two forms: (1) the physical absence but the psychological presence of loved ones and (2) the physical presence but the psychological absence of loved ones [26]. These losses are particularly challenging to address due to their ambiguous nature. These unresolved relational issues also caused complicated family stress and anger outbursts that usually led to domestic violence especially during the early stage of resettlement.

In the community subsystem, the initial support from the resettled countries defines how fast refugee communities recover from adversity and cumulative traumatic stress [21,25]. Local resettlement infrastructures systematically fail to provide mental health support to newly resettled refugees [20]. Cambodian refugees saved from the genocide were not provided adequate food, housing, language, or transportation support at their arrival [15,27]. The need to survive in the U.S. forced them to continue harsh labor jobs and keep their psychological trauma in a frozen state [9,11]. Table 1 is the summary of the literature review on individual mental health and relational consequences of trauma exposure among Cambodian refugees in the U.S. since their initial resettlement.

To date, there have been very limited culturally relevant interventions developed to address the mental health and relational consequences of genocidal trauma exposure and resettlement stress within the Cambodian communities across the U.S. Some notable examples of cultural adapted interventions aiming to address the complexities of prolonged trauma exposure experienced by Cambodian refugee families in the U.S. are the “Cultural Bereavement” [41] and the “Somatic Focused CBT” [42] models. Even though these models were developed in the early 1990s and 2000s, they have not been broadly disseminated and tested for their effectiveness beyond the researchers’ clinics. Additionally, because multiple scholars continued to document that intergenerational transmission of psychopathology and relational disruptions exist in high prevalence among Cambodian refugee families [19,34,43,44], a formal needs assessment of this population after four decades of resettlement in the U.S. is a necessary foundation for future clinical research projects on culturally relevant intervention development, dissemination, and implementation to support Cambodian refugee communities healing post-resettlement. Therefore, this is the first and most updated study that aimed to formally document the mental health and relational needs of Cambodian refugee families across the U.S.

## 2. Materials and Methods

### 2.1. Research Design

This study employed principles of Critical Ethnography [45] to document Cambodian refugees’ mental health and relational needs. In-depth interviews were conducted, transcribed, and coded in Khmer or/and English. Interviews started with the grand tour questions, “Could you share what you see as the overall adjustment and well-being of Cambodian families who resettled your area?” Mini-tour questions center around family relationships, parent–child relationships, couple relationships, children and adolescents, mental health, substance use, war-related experiences, broader socio-political community needs, and professional environments.

### 2.2. Participants and Sites

Eighteen professionals serving Cambodian families across the U.S. (i.e., California, Massachusetts, Pennsylvania, Minnesota, Washington, and Connecticut) were interviewed virtually from May to August 2022. Four participants represent the Cambodian grandparent generation, eleven represent the Cambodian parent generation, and three represent expatriates who are not Cambodian born and who understand Cambodian refugees across migration stages. The standard profiles of the participants are listed in Table 2. Specific information was changed between participants to protect confidentiality.

### 2.3. Procedures

After the study researchers received approval from the institutional research board, professionals who met the inclusion criteria were emailed to introduce the study and request for their participation. An informed consent was sent and signed before the interview took place. Interviews were audio recorded, but personal identifiers were removed to ensure confidentiality. Recruitment was guided by respondent-driven sampling. Eighteen informants agreed to participate in the study during four months of data collection (May–August 2022). At the conclusion, a modest incentive was provided.

### 2.4. Data Analysis

Data were collected and analyzed iteratively. Ethnographic data were gathered through observations and informal conversations with key informants via phone, text messages, and emails throughout initial contact with key informants during the recruitment process. The use of multiple reports, key informants, and multiple data sources further expanded our understanding of mental health and relational needs of Cambodian refugee families. The researcher’s reflexivity during recruitment and data collection were also recorded in memos to inform the analysis and for data verification.

All interviews were analyzed using Developmental Research Sequence [46] to articulate cultural knowledge shared by local experts. The DRS was designed with a twelve-step method of conducting ethnographic interviews and qualitative research data analysis. The Twelve Steps of the Developmental Research Sequence is demonstrated in Table 3.

### 2.5. Trustworthiness

Ensuring trustworthiness is crucial in the qualitative research process so that the analysis and results reflect the realities of participants as closely as possible. According to Lincoln and Guba (1985) [47], trustworthiness and ethical standards in qualitative research can be ensured by following four key criteria: (1) credibility; (2) dependability; (3) confirmability; and (4) transferability. All criteria were followed with thick descriptions and auditing processes.

### 2.6. Ethical Considerations

Fifteen out of eighteen participants were both professionals and former Cambodian refugees themselves. Participants were reminded that they could withdraw their participation at any time during the interview if they became distressed and uncomfortable. At the end of the interview, participants were debriefed. None of the participants withdrew their participation, but instead they shared that they were happy to express their experience and perspectives to inform the mainstream culture about the needs of their underserved communities across the U.S.

## 3. Results

Three domains across migration periods (i.e., Pre-Migration, During Migration, and Post-Resettlement) were elaborated in the larger study. The focus of this manuscript is to describe findings related to Post-Resettlement Experiences in the U.S. (Figure 1). Individual quotes are represented below by using P to refer to the participant number, E to refer to an expatriate who is a non-Cambodian professional, and G to refer their representative generation, from first to third.

### 3.1. Post-Resettlement Experiences in the United States

#### 3.1.1. Category: Impact on Self

The theme **psychological trauma** was reported across generations at post-resettlement. The grandparent generation continues to suffer from mental disorders (i.e., PTSD, depression, anxiety attacks, hallucinations, suicide ideation/attempts, complicated grief, acculturation stress, etc.) due to their prolonged exposure to genocide trauma, refugee camps, and post-resettlement stress without culturally relevant support. Participant P1G2 shared:


*“I feel so sorry for our elders. They could not do much for themselves due to their untreated trauma and physical illnesses. Most of them are at home and experience extreme boredom, hopelessness, and sadness. You know, in America, when you cannot drive nor speak the language and you are old, there’s not much you can do. This makes their overall health even worse. Sometimes, their children and grandchildren don’t respect them because the elders depend too much on the young. This leads to anger, resentment, shame, and all kinds of stressful emotions.”*


On the other hand, the parent generation whose childhoods were severely disrupted during the genocide and life in refugee camps presented behavioral issues (i.e., Oppositional Defiant Disorder and Attention Deficit Hyperactive Disorder) along with mood disorders (i.e., anxiety disorders, anger outburts, depression) during initial resettlement as reported by some informants who work in school settings. Participant P10G2 shared:


*“In the 1990s and early 2000s, many youths disappeared from school, sneaked out at night, and had school problems. These youth were referred to us, and we diagnosed them with ODD, ADHD, adjustment disorder, anxiety disorder, and depression. After years of experience and observation, I noticed that these youth were from very dysfunctional families where their parents lived with severe mental illnesses. There’s no doubt they acted out at home and school.”*


The theme **combined maladaptive and adaptive coping strategies** were reported by all informants. Maladaptive coping includes gambling, excessive drinking, and abusing substances in the grandparent and parent generations as self-medication of mental health complications post-resettlement. It was also reported that these maladaptive coping strategies somehow protect them from committing suicide. Participant P15G2 shared:


*“Gambling, drinking, substance abuse, and homelessness are very common in [the] Cambodian community. Elders use gambling and drinking to cope when their mental health is severely triggered. The younger generation drink and abuse substances when they are stressed. Those who are homeless are usually overdosed on substances, live with mental illnesses, and are in their 20s or 30s.”*


On the other hand, adaptive coping strategies, such as participating in community gardening and cooking, volunteering at Buddhist temples, joining meditation groups, and exercising were also reported. Engaging in the community, culture, and food has always been a part of the healing among Cambodian refugees post-resettlement. Participant P18G2 shared:


*“In our organization, we created community gardening and exercise groups for our elders. We give them taxi vouchers to come to join the activities. They enjoy coming together and do[ing] gardening and exercising together. We believe that these support groups help our elders.”*


#### 3.1.2. Category: Couple Relationships

Cambodian traditional couple relationships changed enormously since initial resettlement to the U.S. In Cambodia, couple relationships follow male-dominant rules. Traditionally, men served as the family head by earning income and making important decisions for the family, while women were homemakers. In contrast, upon their arrival in the U.S., many men experienced family role crises as women also worked outside the home. It was a struggle for men to accept that they now had to share housework duties previously assigned to women. 

The theme **gender role tension** emerged in the responses of most informants. This gender role tension happened due to the family role crisis along with the lingering effects of genocide trauma without professional support during the initial resettlement. Many women assumed men’s roles of generating income to ensure their family survival post-resettlement, while men had to assume women’s roles as homemakers. Unfortunately, many couples could not process these gender tensions, leading to avoidance and passive-aggressive communication in their couple relationship. Participant P12G1 shared:


*“You know what? When Cambodians arrived here, both men and women had to work outside the house to pay the bills. They experience[d] the dramatic change of traditional gender roles, but many men could not accept it at first. That caused so much tension in their relationship.”*


These gender role tensions also led to other consequences. First, infidelity occurred because men deemed themselves useless in their family. This also led to many older men returning to Cambodia and marrying a younger wife. Second, domestic violence also occurred because anger took over when men could not process their role crisis and their cumulative traumatic stressors. Participants who were domestic violence case workers reported that domestic violence occurred regularly during initial resettlement but decreased in later stages. Third, divorce occurred when couples recognized high levels of relational tension and stress. Many younger women filed for divorce and entered into interracial marriages, while older women chose to remain in their marriages despite their discontent. 

Therefore, the theme **need for couple and family therapy** also emerged. Informants emphasized that Cambodian couples do not go to therapy when they have couple conflicts due to the stigma of not sharing family problems outside of the home. Only when the problems become severe do they ask for help, which often happens in prison or court for domestic violence and divorce cases.

#### 3.1.3. Category: Parent–Child Relationships

The theme **parenting disrupted by war and migration** was reported by most informants. Cambodian refugee parents experienced barriers to effective parenting due to their low educational level, socioeconomic status, and mental health status compromised by the war and migration. These parents struggled to acculturate and could not implement effective parenting in another culture with very limited culturally responsive and trauma-informed support available to them. Therefore, Cambodian parenting was severely disrupted and had a limited chance for improvement without professional assistance (i.e., parenting-focused support or programs). Participant P12G1 shared:


*“The most important factor that defines the success of refugee children is their parents. According to my observation, about half of these children made it, while half did not make it in terms of adjustment. Those who made it had parents who had some education and could pick up English and employment skills quickly. In contrast, those who could not make it had parents who were on disabilities–either mentally or/and physically–and had no education. If parents received enough support at the initial resettlement, children and the whole family would have benefited from it, too.”*


The grandparent and parent generations also cited incidents of parent–child relationship breakups due to the barriers erected by ineffective parenting. Some children decided to move out and others completely cut ties with their parents due to their parents’ inability to carry out parental responsibilities, deemed as second-level family separation. Participant P15G2 shared:


*“Cambodians experienced painful losses and family separation, so their wish is always for family reunions. However, a lot of children decide to move away from their parents because there’s so much tension and anxiety in their relationships. Children said they must leave home for their mental health. This added to another layer of pain and heartbreak for refugee parents who have already suffered so much from family separations.”*


The theme **need for effective parenting practices** was reported by all informants. Current parents reported that they need effective parenting skills and sustainable parent support groups. Unlike the grandparent generation, many of these younger parents live in non-traditional family compositions, such as same-sex and interracial households. These respondents stated a need for trauma-informed, interracial, and culturally responsive parenting skills in the American context. Participant P11G2 shared:


*“I’m in an interracial marriage. My husband is White American, and we have a teenage daughter. Our parenting goal is to make sure that our daughter is successful in America. To be honest, I’m very concerned about the loss of Khmerness in my daughter. I notice that she is more interested in being American than being Khmer, while she is also aware that she is raised by a Khmer mother. She is very disconnected from her grandmother too as they have different intergenerational interests and values. There are many parents my age who are in [an] interracial marriage and face similar concern[s].”*


#### 3.1.4. Category: Context

The theme **mental health stigma** remains common among Cambodian families, according to most informants, due to a lack of mental health awareness and emotional regulation skills in the Cambodian communities. Depression and anxiety symptoms are often labeled as being lazy or impulsive. Moreover, when a family member receives a diagnosis of mental illness, that individual tends to remain silent. Some families ignore their mental health until it moves to the crisis stage before they seek support. Participant P16E shared:


*“There’s still [a] stigma of mental health among [the] Cambodian community. You may notice the first encounter when they seek support is in the emergency room, police station or jail. They leave their problems there until they become very severe before they seek help.”*


The theme **lack of culturally tailored services** emerged in the responses of all the participants when asked about the offered services specifically structured to meet the needs of Cambodian refugee families. In other words, the informants stated that many services remain inaccessible to the Cambodian refugee population due to a lack of understanding and cultural knowledge of service providers. On a broader societal level, the data shows that services related to mental health, family relationships, education, and legal consultation in the case of deportation have not been tailored for this specific population. Some nonprofit organizations employ staff who strive to design programs for elders and youth, yet leave couples-focused and family-focused services off the list due to a lack of resources. In summary, the barriers to access persist which leads to health disparities among Cambodian refugees. Participant P11G2 shared:


*“Trauma-informed care and cultural sensitivity need to be introduced to professionals in legal, education, and mental health fields. Those who do law enforcement should consider the history of Cambodian genocidal trauma. Clinicians should know their client’s history of resettlement, culture, and values. These professionals should learn a different unique way [not the standard one] when working with ethnic minority populations, such as Cambodian families.”*


## 4. Discussion

Trauma exposures during the genocide, refugee camps, and post-resettlement continue to impact Cambodian refugee communities four decades after resettlement. Based on the findings of this needs assessment, individual mental health and relational issues become more and more complicated over time without much public and scholarly attention.

### 4.1. Impact on Individuals

#### 4.1.1. Mental Health Crisis

Key informants reported that PTSD, depression, anxiety disorders, and a host of mental health complications continue to plague Cambodian refugees, especially those who live in highly disrupted families. Mental illnesses were also reported across generations, but most severely among the grandparent and parent generations. On the other hand, the grandchildren generation born post-resettlement commonly battle depression, anxiety disorders, and behavioral problems. Cambodian refugee communities across the U.S. failed to receive sufficient attention over the years. Specifically, many grandparents who were severely affected by the killing fields were reported to battle mental illnesses and comorbid conditions and have been dying without culturally responsive treatments. Existing literature regarding war-affected refugees further attests to the prevalence of mental health crises among this population, which is compounded by acculturation stressors post-resettlement as resettlement countries often take refugee’s mental health for granted [5,20,48]. Therefore, newly resettled refugees often remain trapped in cycles of psychopathology such as PTSD, adjustment disorder, and complicated grief for generations with limited hope of recovery without culturally responsive multi-level interventions [49].

#### 4.1.2. Overall Health Crisis

Cambodian refugees in the biggest cities across the U.S. reported poor overall health due to comorbid conditions [50]. This study highlighted the unaddressed comorbidity of mental and physical illnesses in Cambodian refugee communities: PTSD, anxiety disorders, high blood pressure, stroke, and diabetes type II were reported with high prevalence. Most informants reported confusion experienced by Cambodians when receiving multiple diagnoses from their health providers. These individuals reported the side effects that go along with taking multiple medications; to solve this problem, they often pause taking the prescription or decrease the dosage without reporting it to their healthcare providers. The findings of this study are consistent with results from previous studies conducted among Cambodian refugees two decades ago highlighting the comorbidity of mental health and physical health conditions [15,22,38]. This means that the overall health conditions of this population have not been addressed effectively. Therefore, it is essential that healthcare providers who work with Cambodian refugees understand the mental and physical health consequences of trauma exposure during the genocide and their current needs to effectively address their health issues.

### 4.2. Impact on Family Relationships

#### 4.2.1. Disrupted Family Roles

Parents who fail to carry out their family roles due to mental and physical illnesses leave their children to navigate life by themselves. Disrupted family relationships led to severe challenges to traditional family roles with refugee communities post-resettlement [51]. Cambodian refugee families experienced family role crisis and role ambiguity during initial resettlement due to the absence and/or death of significant family members during the genocide. Oftentimes, older children assumed leading roles to keep the family functioning. However, not many families successfully adjusted because parents struggled to acculturate, while children could not effectively take on these leading roles and responsibilities. This study further affirms that refugee minors whose parents could not perform their roles experienced school and relational problems both at home and at school [18,27,51]. Therefore, in disrupted family systems, both parents and children need support to successfully acculturate in a foreign setting. The mental and relational impact of growing up in a disrupted family persists in Cambodian refugee communities.

#### 4.2.2. Disrupted Gender Roles

During initial resettlement, many Cambodian couples complied with the traditional gender roles in which men took the leading role of generating family income. This status quo encountered challenges and broke down in the U.S. as women also worked outside of the home to ensure the survival of the family. This meant that women expected more equalitarian partnerships within the housework as they were also part of the paid labor force. Many men refused to accept this gender role change, causing severe relational tensions. Results from this study emphasized that older couples tended to stay in the marriage, communicate using passive-aggressive styles, and separated emotionally from each other; this sometimes led to infidelity, and older men returned to Cambodia to marry younger women. Conversely, younger couples made attempts to communicate and strive for more equal partnerships, which resulted in high divorce rates and interracial re-marrying. The findings of this needs assessment are aligned with previous studies conducted among Cambodian communities that underscore high family stress, infidelity, financial stress, aggression [27], parenting stress, and children’s school problems [38]. 

#### 4.2.3. Intergenerational Transmission of Trauma

The long-term consequences of trauma exposures on mental health and family relationships in refugee communities are not typically interrupted without culturally responsive multi component interventions [49,52]. Participants in this study reported that grandparent, parent, and grandchildren generations still struggle with similar mental illnesses, relational problems, and maladaptive ways of coping in the absence of receiving effective mental health treatment. Therefore, reports from multiple informant perspectives who work in various professional settings support the existing literature regarding the mental health and relational consequences of war-related trauma exposures and forced displacement [21,53,54]. To date, Cambodian refugee families, especially those experiencing severe family disruptions, remain trapped in cycles of intergenerational transmission of trauma and relational disruptions [19,34].

### 4.3. Lack of Culturally Tailored Services

War-affected refugees who resettled in the U.S. have different migration histories, cultures, and various health conditions [55,56]. Service providers often lack the training to provide culturally responsive care to specific ethnic minority refugees, and the larger social system does not offer enough resources to address health disparities among refugee populations [55]. The existing literature also pointed to cultural, language, and financial barriers that prevent Cambodian refugees from seeking mental health services despite their consistently high rates of PTSD and major depressive disorders [57,58]. Cultural beliefs and stigma related to mental health conditions are also key barriers to accessing services among refugee populations [52]. The lack of culturally tailored services exacerbates these barriers, leading to a refugee health crises and persistent intergenerational transmission of ill health. Some nonprofit organizations in the Cambodian refugee communities strive within their capacity to respond to the needs of elders, youth, and parents. However, they are still missing access to culturally responsive services in most resettlement states of the U.S. in addressing the complexities of current mental health and relational needs of Cambodian refugee communities. In the larger U.S. society and global context, few initiatives exist that tailor services—particularly at the family and community level—to specific ethnic minority refugee populations [49,55,56], and that includes Cambodian refugees. Specifically, models on Cultural Bereavement [41] and Somatic Focused CBT [42] were developed in the early 1990s and 2000s with specific aims to address psychological trauma and mental health complications among Southeast Asian refugees, including Cambodian refugees post-resettlement. Unfortunately, these models have not been broadly disseminated or tested for feasibility and effectiveness within these refugee communities. Barriers to testing these approaches could be related to the lack of local human resources (e.g., trainings related to trauma-informed care and culturally responsive approaches, advanced provider skills to work with refugees affected by forced displacement and war trauma, and language and cultural differences between providers and refugees) and the lack of funding mechanisms to support model development and testing specifically tailored to ethnic minority refugee populations.

### 4.4. Implications and Future Directions

#### 4.4.1. Clinical Implications

The current needs assessment among Cambodian refugee families in the U.S. revealed significant ongoing deficiencies in terms of mental health and relational services among the population. Key informants who were former refugees themselves and have worked with their own population across migration stages agreed that the Cambodian refugee communities across the U.S. reflect an ongoing need for culturally tailored systemic interventions. 

Individual-based interventions could address Cambodian refugees’ mental disorders, especially for the grandparents and the parents who were exposed to severe traumas during the war and refugee camps. PTSD treatment and strength-based systemic approaches in combination with trauma-informed care are needed to support Cambodian refugees. 

Systemic family therapy in combination with trauma-informed care and family-based interventions focusing on promoting couple relationship and positive parenting practices for different types of Cambodian contemporary family systems (i.e., single-mother households, interracial households, and same-sex parent households) are needed to ensure the successful development of younger Cambodian-Americans. Importantly, Cambodian refugees in nontraditional family structures need unique support to navigate the communication of value systems between cultures. Experiencing multiple nonnormative life transitions, Cambodian refugee parents need formal spaces to reflect and revise the past, current, and future parenting in the context of forced displacement and post-resettlement.

At the community level, having a home-like environment in a foreign setting is significant for trauma-affected and displaced populations. In this study, informants reported that Cambodians feel peace and resolve some grief by visiting Buddhist temples where they can connect with their people, culture, food, and language. Some cultural events organized by Buddhist temples help bring the Cambodian community together, but a need still exists for more sustainable community-based interventions to remove stigma and to bring awareness of mental health after trauma exposures and to promote healthy family relationships post-resettlement.

#### 4.4.2. Research Implications

The objective of the current needs assessment was to address gaps—such as the lack of culturally responsive family-based and community-based interventions—in the refugee family literature. Therefore, key findings in this study support the development of feasibility studies to adapt evidence-based family- and community-based interventions to address the lack of culturally responsive multi-level interventions for mental health and relational impacts of Cambodian refugee families in the U.S. Moreover, the barriers of implementation and dissemination of the interventions should be addressed so that the Cambodian refugee communities across the U.S. can effectively benefit from them.

Next, multi-method approaches (i.e., qualitative, quantitative, and mixed methods) should be integrated with the golden standard randomized control trials (RCTs) in the context of Cambodian refugee families who still struggle with various aspects of their lives (i.e., mental health, acculturation, education, employment, finances, and family relationships). The integration of qualitative methods, such as Community Participatory Research, Ethnography, Photovoice, and Phenomenology, are essential to incorporating local cultural voices and empowering community members to advocate for their families at the policy level. Postmodern paradigmatic approaches and critical methodological frameworks have demonstrated success in supporting prevention and intervention efforts with specific attention to translational sciences that inform effective and sustainable dissemination and implementation strategies.

## 5. Conclusions

Cambodian refugee communities in the U.S. endure the mental health and relational consequences of multiple trauma exposures, family separation, unresolved grief, disruptions to personal and family development, and limited social support systems that go on uninterrupted for over four decades of their resettlement. Because many adversities that these refugees experience post-resettlement could have been prevented, this study serves as another call—after multiple calls—for action to implement prevention science at all system levels. Particularly, PTSD and other trauma related disorders should be immediately addressed at the individual level using culturally tailored and trauma-informed care frameworks, while relational ruptures resulting from family separations and death during the genocide and migration should be intervened by using culturally responsive and trauma-informed family-based and community-based programs to promote healing in the aftermath of trauma exposure and displacement. The current needs assessment highlighted ongoing relational and mental health disparities within Cambodian refugee communities in the U.S. This is a critical social justice and human right issue as Cambodian refugee families have the right to access health, social, and protective services to support themselves and their families to work through past and current adversities. 

## Figures and Tables

**Figure 1 behavsci-14-00535-f001:**
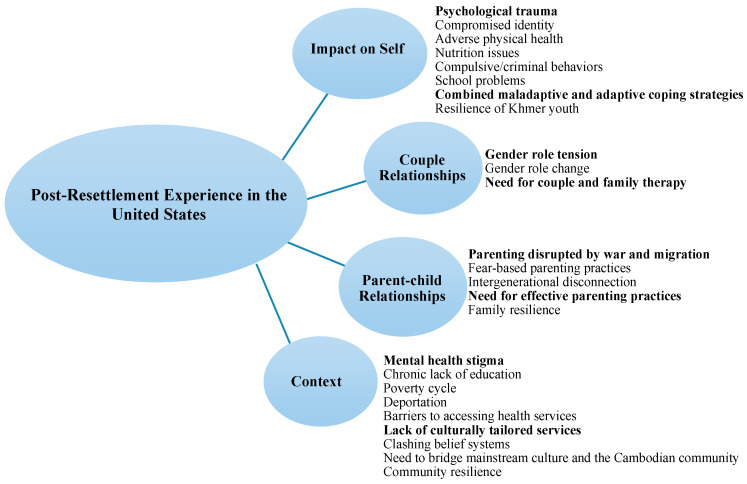
Post-Resettlement Experience in the United States.

**Table 1 behavsci-14-00535-t001:** Summary of the literature of Cambodian refugees at initial resettlement.

Individual Subsystem		
Studies	Population and Site	Findings
** *PTSD Among Cambodian Refugees* **		
[23]	Indochina refugees (Vietnamese, Laotian, Cambodian)	-Cambodian refugees were the highest traumatized (10 trauma events and 2 torture experiences). -Comorbidity of mental and physical illnesses was found.
[22]	Cambodian refugees	-A mean of 14 genocidal traumatic events and 1.3 post-resettlement trauma exposures was revealed. -A strong relationship between trauma exposures and PTSD, depression, dissociation, and cultural syndromes was found.
[27]	Cambodian refugee women in CA and MA	-Somatic symptoms of PTSD (i.e., chest pain, heart palpitation, shortness of breath, dizziness, etc.) were reported.
[28]	Cambodian refugee women in the U.S. and France	-A high prevalence of depression, anxiety, and ruminating of past genocide trauma were reported.
[29]	Cambodian refugees with PTSD diagnosis in MA	-A high prevalence of the cultural syndrome “Thinking a lot” was reported.
[30,31]	Cambodian refugees in the U.S.	-A high prevalence of severe mental ailments (i.e., PTSD, dissociation, depression, anxiety disorders) was surprisingly found among non-clinical samples.
[32]	586 Cambodian refugees in CA	-Severe genocide trauma exposures (i.e., near-death starvation and murder of family members) were reported.
** *Substance Abuse* **		
[33]	Cambodian refugees	-Substance abuse in response to mental illnesses was reported.
[34]	Cambodian refugees in the southern coast of the U.S.	-Excessive alcohol and tobacco use by other family members were reported.
[32,35]	339 Cambodian refugees in the largest community in the U.S.	-Alcohol abuse of others was reported, not of their own.
** *Comorbid Conditions* **		
[22]	Cambodian refugees	-The comorbidity of major affective disorders, PTSD, medical, and social disabilities were found.
[15]	136 Cambodian refugees in CT and MA	-The comorbidity of depression, PTSD, and physical illnesses (i.e., diabetes, hypertension, and stroke) in all age groups, and premature death were reported.
[36]	100 Cambodian refugees in CT	-The association between social isolation and poor mental and physical health was found.
[37]	Khmer refugees in MA	-The correlation between psychiatric and physical illnesses was found.
[38]	Khmer refugees in MA	-The association between “worry attacks” and PTSD was found.
**Family Subsystem**		
** *Family Stress* **		
[27]	60 Cambodian refugees in CA and MA	-Aggression toward others and social isolation were reported.
[38]	Cambodian clinical sample	-Family stressors (i.e., financial stress, children’s school problems, and concerns about relatives in Cambodia) were associated with “worry attack”.
** *Anger Outbursts* **		
[39]	Cambodian refugees in MA	-Anger outbursts were found to be related to tension in marital relationships through verbal and physical violence.
**Community Subsystem**		
** *Community Issues and Lack of Support System* **		
[18,40]	Cambodian refugees	-Gang violence, aggressive behaviors, school dropout, substance abuse, emotional dysregulation, and mental illnesses that led to incarceration and deportation were reported.

**Table 2 behavsci-14-00535-t002:** Participant demographic information.

Participant	Age	Gender	Profession	Work Setting
**P1G2**	35–55+	Male	Psychologist	Various Settings
**P2G1**	60–70+	Male	Director	Non-Profit Organization
**P3G2**	35–55+	Non-binary	Director	Non-Profit Organization
**P4G2**	35–55+	Female	Social Worker	Prison
**P5G2**	35–55+	Male	Psychologist	University
**P6G2**	35–55+	Female	Psychologist	Prison
**P7E**	35–55+	Female	Professor	University
**P8G2**	35–55+	Male	Researcher	Non-Profit Organization
**P9G1**	60–70+	Female	Case Manager	Non-Profit Organization
**P10G2**	35–55+	Male	Social Worker	Non-Profit Organization
**P11G2**	35–55+	Female	Director	Non-Profit Organization
**P12G1**	60–70+	Female	Case Manager	Non-Profit Organization
**P13G2**	35–55+	Female	Therapist	Non-Profit Organization
**P14E**	35–55+	Female	Medical Anthropologist	Non-Profit Organization
**P15G2**	35–55+	Female	Program Manager	Non-Profit Organization
**P16E**	35–55+	Male	Psychologist	Non-Profit Organization
**P17G1**	60–70+	Female	Case Manager	Non-Profit Organization
**P18G2**	35–55+	Male	Program Manager	Hospital

***Note.* P** = Participant; **G1** = First generation/Grandparent generation; **G2** = Second generation/Parent generation; **E** = Expatriate.

**Table 3 behavsci-14-00535-t003:** Twelve Steps of the Developmental Research Sequence [46].

Step	Development Research Sequence
**1**	Locating an informant
**2**	Interviewing an informant
**3**	Making ethnographic record
**4**	Asking descriptive questions
**5**	Analyzing ethnographic interviews
**6**	Making a domain analysis
**7**	Asking structural questions
**8**	Making a taxonomic analysis
**9**	Asking contrast questions
**10**	Making a componential analysis
**11**	Discovering cultural themes
**12**	Writing an ethnography

## Data Availability

The data generated and/or analyzed during the current study are not publicly available nor are they available on request due to privacy and ethical concerns.

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
