# Peer review of "Mental Health and Relational Needs of Cambodian Refugees after Four Decades of Resettlement in the United States: An Ethnographic Needs Assessment"

_behavsci, 2024, doi:10.3390/bs14070535_

Round 1

Reviewer 1 Report

Comments and Suggestions for Authors

This is a very well written manuscript that describes a timely and important topic - a formal needs assessment of the Cambodian refugee population after four decades of resettlement in the U.S. The study collected qualitative data from ethnographic interviews conducted virtually from May to August 2022 with 18 professionals serving Cambodian families across the U.S. (California, Massachusetts, Pennsylvania, Minnesota, Washington, and Connecticut). Data was analyzed using 12-steps Developmental Research Sequence. The authors describe the themes that emerged from the analyses.

In the Introduction, the authors present a compelling argument for the need to reevaluate the problems that continue to afflict the Cambodian communities in the U.S. Unfortunately, the impact of intergenerational trauma has been documented in several vulnerable sub-groups in the U.S. and probably not better addressed. It is therefore suggested that the authors modify their conclusions to include recommendations more specific to the Cambodian community rather than a general advocacy for basic rights (“Refugee families have the right to have access to health, social, and protection services to support themselves and their families to work through adversities”). In addition, there is extensive literature on culturally specific interventions for Cambodian refugees that the authors may wish to mention (e.g., M. Eisenbruch’s concept of cultural bereavement; DE Hinton’s somatically focused CBT). It is interesting to understand whether this body of knowledge has not been disseminated or made available to refugees beyond the specific clinics that were actively involved with the community.    

Specific suggestions:

-The paper should be shortened; the issues raised in Discussion and Implications seem to be repeating similar language to that of the findings and are using very general ideas. For example: "Cambodian refugee parents need formal spaces to reflect and revise the past, current, and future parenting in the context of forced displacement and post-resettlement." 

-The authors may consider summarizing the information in Table 2 in a short paragraph.

In addition to the comments above, I wish to express my concern that while the topic is important and the work described is original and of good quality, there is not much new knowledge that can be drawn from their findings. Their recommendations seem to be relevant to any vulnerable group in the U.S. I would urge them to be more specific to the resettled Cambodian refugee population. In addition, the manuscript should be shortened.

I believe it should be published but following meaningful revisions.

Author Response

May 25th, 2024

Dear Drs. Kotaro Shoji and Andrew Smith,

Thank you for considering this manuscript for publication in the special issue “Trauma, Resilience and Mental Health”of Behavioral Sciences. We reviewed the comments carefully and attempted to be responsive to the thoughtful suggestions and concerns. Please find a description of how we addressed each comment below.

Reviewer #1

Open Review                        (x) I would not like to sign my review report  

(  ) I would like to sign my review report  

Quality of English Language         ( ) I am not qualified to assess the quality of English in this paper  

( ) English very difficult to understand/incomprehensible  
( ) Extensive editing of English language required  
( ) Moderate editing of English language required  
( ) Minor editing of English language required  
(x) English language fine. No issues detected 

Yes

Can be improved

Must be improved

No applicable

Is the content succinctly described and contextualized with respect to previous and present theoretical background and empirical research (if applicable) on the topic?

( )

(x)

( )

( )

Are the research design, questions, hypotheses and methods clearly stated?

(x)

( )

( )

( )

Are the arguments and discussion of findings coherent, balanced and compelling?

( )

(x)

( )

( )

For empirical research, are the results clearly presented?

(x)

( )

( )

( )

Is the article adequately referenced?

(x)

( )

( )

( )

Are the conclusions thoroughly supported by the results presented in the article or referenced in secondary literature?

( )

(x)

( )

( )

Comments and Suggestions for Authors

This is a very well written manuscript that describes a timely and important topic - a formal needs assessment of the Cambodian refugee population after four decades of resettlement in the U.S. The study collected qualitative data from ethnographic interviews conducted virtually from May to August 2022 with 18 professionals serving Cambodian families across the U.S. (California, Massachusetts, Pennsylvania, Minnesota, Washington, and Connecticut). Data was analyzed using 12-steps Developmental Research Sequence. The authors describe the themes that emerged from the analyses.

RESPONSE: We thank reviewer #1 for carefully reading through our manuscript and thoughtfully giving us feedback and recommendations that will help us to polish our manuscript and make it more impactful.

In the Introduction, the authors present a compelling argument for the need to reevaluate the problems that continue to afflict the Cambodian communities in the U.S. Unfortunately, the impact of intergenerational trauma has been documented in several vulnerable sub-groups in the U.S. and probably not better addressed. It is therefore suggested that the authors modify their conclusions to include recommendations more specific to the Cambodian community rather than a general advocacy for basic rights (“Refugee families have the right to have access to health, social, and protection services to support themselves and their families to work through adversities”). In addition, there is extensive literature on culturally specific interventions for Cambodian refugees that the authors may wish to mention (e.g., M. Eisenbruch’s concept of cultural bereavement; DE Hinton’s somatically focused CBT). It is interesting to understand whether this body of knowledge has not been disseminated or made available to refugees beyond the specific clinics that were actively involved with the community.    

RESPONSE: Thank you for your comments regarding concluding remarks of this manuscript. We agree that the conclusion should be specifically tailored to the Cambodian refugee communities that were the target of the study. We made modifications to the conclusion section consistent with this request.

Thank you for highlighting Dr. Maurice Eisenbruch’s work on cultural bereavement. We made the initial decision not to include the literature examining Cambodian and Southeast Asian refugees’ experiences of bereavement because we believe that content is beyond the scope of this manuscript. Instead, we used the human ecological framework to guide our systemic focus in this ethnographic needs assessment documenting mental health and relational needs of Cambodian refugees across system levels (i.e., individual, family, and community). If we include a review on bereavement we would also incumber the task of referencing additional related work impacting refugee mental health.

We are also grateful for the nod to Dr. Devon Hinton’s research on Somatically Focused CBT, which includes a comprehensive concept paper that elaborates on the process of cultural adaptation of CBT for Cambodian refugees. We made the decision not to reference it here because to our knowledge the model has not been tested to demonstrate feasibility or effectiveness for implementation with Cambodian refugee communities in the U.S. This is important work that we hope will be carried out and tested in the future. We will contact Dr. Hinton and research team to learn more about their model for potential collaboration.

Specific suggestions:

-The paper should be shortened; the issues raised in Discussion and Implications seem to be repeating similar language to that of the findings and are using very general ideas. For example: "Cambodian refugee parents need formal spaces to reflect and revise the past, current, and future parenting in the context of forced displacement and post-resettlement." 

RESPONSE: Thank you for your suggesting that we consider removing redundancy and shortening the manuscript. We were purposeful about including extensive information in various sections as this manuscript represents the first formal ethnographic mental health and relational needs assessment ever conducted among Cambodian refugee communities in the U.S.

We agree that the language we used in the discussion and implications sections has some redundancies. We revised to remove repetition when possible without losing our ability to make important links. We also added implications specific to addressing the current needs of Cambodian refugees across system levels.

-The authors may consider summarizing the information in Table 2 in a short paragraph.

RESPONSE: Thank you for suggesting that we summarize the information in Table 2 in a short paragraph. Because another reviewer requested more details in Table 2 to verify quotes presented in the results section, we believe that Table 2 is still useful to keep, even though we agree with that it could be replaced by a paragraph.

In addition to the comments above, I wish to express my concern that while the topic is important and the work described is original and of good quality, there is not much new knowledge that can be drawn from their findings. Their recommendations seem to be relevant to any vulnerable group in the U.S. I would urge them to be more specific to the resettled Cambodian refugee population. In addition, the manuscript should be shortened.

RESPONSE: We agree that much of the information generated in this needs assessment has been found to be present across other refugee populations. Yet, it is alarming to us that the mental health issues (i.e., PTSD and other trauma related disorders) and relational disruptions (i.e., parent-child, couple, and intergenerational relationship ruptures) are still highly prevalent and uninterrupted after 40+ years of the Cambodian refugee resettlement to the U.S. This implies that this population has been neglected and underserved. Indeed, like the case with various refugee communities, this represents a human rights and social justice issue. We made efforts to shorten the manuscript but also believe that it is necessary to keep much of the information for clarity and comprehension purposes.

I believe it should be published but following meaningful revisions. 

RESPONSE: Thank you for believing that this manuscript is important and should be published.

Submission Date

28 April 2024

Date of this review

17 May 2024 21:16:33

Reviewer 2 Report

Comments and Suggestions for Authors

Dear authors,

thank you for this interesting paper, which is very concise and comprehensive.

I have two remarks. The first one concerning the validity of the selection of the quote for the presentation of the results: I do have the feeling that the quotes regarding the topics couple relationship are coming from female interviewees (this is a very personal interpretation of the chosen quotes as a male researcher). This might influence the interpretation of the impact of trauma on the couple relations. Maybe you could present the gender of the interviewees in table 2, to make it possible to follow your interpretation better. (And in case all quotes are really coming from one gender either present additional quotes from other genders or point to a possible bias.) Aletrnatively, I would suggest one sentence concerning the limitations of the research regarding sampling (as I understand all interviewees had Cambodian background - maybe experts with other ethnic background would have different insights, with all the connected challenges...).

The other concern is related to the suggested implications. I wonder if the case of Cambodian refugees is so different to refugees from other parts in the world or even to other migrants (not refugees but maybe illegal migration) regarding e.g., changes in gender roles, acculturation stress. So my point is if there is comparable data on other special groups or even the mainstream population pointing to the special case of Cambodian refugees and there offspring or to the possiblity of generalization for more/different groups?

I hope this feedback helps

all the best

Author Response

May 25th, 2024

Dear Drs. Kotaro Shoji and Andrew Smith,

Thank you for considering this manuscript for publication in the special issue “Trauma, Resilience and Mental Health”of Behavioral Sciences. We reviewed the comments carefully and attempted to be responsive to the thoughtful suggestions and concerns raised by the reviewers. Please find a description of how we addressed each comment below.

Reviewer #2

Open Review                       (x) I would not like to sign my review report  

(  ) I would like to sign my review report  

Quality of English Language          ( ) I am not qualified to assess the quality of English in this paper  

( ) English very difficult to understand/incomprehensible  
( ) Extensive editing of English language required  
( ) Moderate editing of English language required  
( ) Minor editing of English language required  
(x) English language fine. No issues detected 

Yes

Can be improved

Must be improved

No applicable

Is the content succinctly described and contextualized with respect to previous and present theoretical background and empirical research (if applicable) on the topic?

(x)

( )

( )

( )

Are the research design, questions, hypotheses and methods clearly stated?

(x)

( )

( )

( )

Are the arguments and discussion of findings coherent, balanced and compelling?

(x)

( )

( )

( )

For empirical research, are the results clearly presented?

(x)

( )

( )

( )

Is the article adequately referenced?

(x)

( )

( )

( )

Are the conclusions thoroughly supported by the results presented in the article or referenced in secondary literature?

(x)

( )

( )

( )

RESPONSE: We thank reviewer #2 for their thoughtful feedback and recommendations that help us to strengthen the manuscript and make it more impactful.

Comments and Suggestions for Authors

Dear authors,

thank you for this interesting paper, which is very concise and comprehensive.

I have two remarks. The first one concerning the validity of the selection of the quote for the presentation of the results: I do have the feeling that the quotes regarding the topics couple relationship are coming from female interviewees (this is a very personal interpretation of the chosen quotes as a male researcher). This might influence the interpretation of the impact of trauma on the couple relations. Maybe you could present the gender of the interviewees in table 2, to make it possible to follow your interpretation better. (And in case all quotes are really coming from one gender either present additional quotes from other genders or point to a possible bias.) Alternatively, I would suggest one sentence concerning the limitations of the research regarding sampling (as I understand all interviewees had Cambodian background - maybe experts with other ethnic background would have different insights, with all the connected challenges...).

RESPONSE: Thank you for your concern regarding gender bias in the quote selection. The quotes are representative of all genders in the sample across themes. All genders expressed similar insights and perspectives regarding gender role changes and tensions within Cambodian refugee communities in the U.S. The decision to only include one representative quote was related to manuscript length. We added gender information in Table 2 to address the concern about gendered representation of results. We did this with careful attention to still protect participant confidentiality by changing other information – changing where they live, age, and profession.

Thank you for suggesting that we add a sentence concerning the limitations of the research with respect to sampling. Three study participants are non-Cambodian born (P7E, P14E, and P16E). We made a modification to make the point clearer. Overall, we believe that we were able to hear both Cambodian and non-Cambodian professional perspectives.

The other concern is related to the suggested implications. I wonder if the case of Cambodian refugees is so different to refugees from other parts in the world or even to other migrants (not refugees but maybe illegal migration) regarding e.g., changes in gender roles, acculturation stress. So my point is if there is comparable data on other special groups or even the mainstream population pointing to the special case of Cambodian refugees and their offspring or to the possibility of generalization for more/different groups?

RESPONSE: Thank you for noting the potential implications of the study and issues around similarities that Cambodian refugees may share with other groups. You make a good point about comparative data across various refugee populations. Indeed, refugee and other displaced populations share commonalities in adverse experiences (as well as resilience factors), but there are also unique experiences related to the reasons for forced displacement, migration and post resettlement conditions. Even though these are important considerations, it is not the focus of this manuscript to compare data across displaced populations. We may make that an area of emphasis in a future manuscript.

I hope this feedback helps

 RESPONSE: We are grateful for your generous and thoughtful observations.

all the best

Round 2

Reviewer 1 Report

Comments and Suggestions for Authors

Thank you for addressing my comments in your reply.

I feel I have to insist on the following:

On line 111-112, the authors state: “To date there has not been any culturally relevant interventions to address the mental health and relational consequences of genocidal trauma exposure and resettlement stress within the Cambodian communities across the U.S.”

I mentioned Dr Eisenbruch’s construct of Cultural Bereacement (year 1990) and Dr. Hinton’s research on Somatically Focused CBT (year 2006) just as examples of culturally specific interventions that were developed and studied many years ago. Given that the authors' conclusions focus on “… the need for culturally tailored multi- level interventions to effectively address mental health and relational challenges of multigenerational Cambodian families”, I found it lacking that the manuscript did not address the culturally tailored treatment methods that were already studied. You could choose to discuss possible reasons that this knowledge has not become more available after 4 decades of resettlement, but to not even mention several of the treatment ideas that were studied in the 1990s and early 2000s specifically regarding the complex mental health needs of the Cambodian refugees resettled in the US, seems to be a problem that should be fixed.

Author Response

June 5th, 2024

Dear Drs. Kotaro Shoji and Andrew Smith,

Thank you for considering this manuscript for publication in the special issue “Trauma, Resilience and Mental Health” of Behavioral Sciences. We reviewed the comments carefully and attempted to be responsive to the thoughtful suggestions and concerns. Please find a description of how we addressed each comment below.

Reviewer #1

Open Review                        (x) I would not like to sign my review report  

(  ) I would like to sign my review report  

Quality of English Language ( ) I am not qualified to assess the quality of English in this paper  

( ) English very difficult to understand/incomprehensible  
( ) Extensive editing of English language required  
( ) Moderate editing of English language required  
( ) Minor editing of English language required  
(x) English language fine. No issues detected 

Yes

Can be improved

Must be improved

No applicable

Is the content succinctly described and contextualized with respect to previous and present theoretical background and empirical research (if applicable) on the topic?

( )

(x)

( )

( )

Are the research design, questions, hypotheses and methods clearly stated?

(x)

()

( )

( )

Are the arguments and discussion of findings coherent, balanced and compelling?

(x)

()

( )

( )

For empirical research, are the results clearly presented?

(x)

( )

( )

( )

Is the article adequately referenced?

(x)

( )

( )

( )

Are the conclusions thoroughly supported by the results presented in the article or referenced in secondary literature?

( )

(x)

( )

( )

Comments and Suggestions for Authors

Thank you for addressing my comments in your reply.

RESPONSE: We thank reviewer #1 for carefully and thoughtfully reading through our revised manuscript. We appreciate that you stressed the significance of mentioning the culturally relevant interventions developed by Dr. Eisenbruch and Dr. Hinton in the early 90s and 2000s even though their interventions were not disseminated beyond their clinics since then. We discussed your recommendation and agreed that it is relevant to include their work, particularly the barriers that prevented their models from becoming available for broader testing and dissemination after decades of the Cambodian refugee resettlement.

I feel I have to insist on the following:

On line 111-112, the authors state: “To date there has not been any culturally relevant interventions to address the mental health and relational consequences of genocidal trauma exposure and resettlement stress within the Cambodian communities across the U.S.”

I mentioned Dr Eisenbruch’s construct of Cultural Bereacement (year 1990) and Dr. Hinton’s research on Somatically Focused CBT (year 2006) just as examples of culturally specific interventions that were developed and studied many years ago. Given that the authors' conclusions focus on “… the need for culturally tailored multi- level interventions to effectively address mental health and relational challenges of multigenerational Cambodian families”, I found it lacking that the manuscript did not address the culturally tailored treatment methods that were already studied. You could choose to discuss possible reasons that this knowledge has not become more available after 4 decades of resettlement, but to not even mention several of the treatment ideas that were studied in the 1990s and early 2000s specifically regarding the complex mental health needs of the Cambodian refugees resettled in the US, seems to be a problem that should be fixed.

RESPONSE: We understand this request and made modifications using track changes in the manuscript. We discussed the possible reasons that the models developed in the 1990s and 2000s have not become more available or successfully implemented and tested with Cambodian refugee communities. Please see the modification on line 113-120, pp. 4 (in the introduction section), and on line 775-785, pp. 13 (in the discussion section: Lack of Culturally Tailored Services).

Submission Date

28 April 2024

Date of this review

03 Jun 2024 10:55:05
